# Effects of the Trp64Arg Polymorphism in the *ADRB3* Gene on Body Composition, Cardiorespiratory Fitness, and Physical Activity in Healthy Adults

**DOI:** 10.3390/genes14081541

**Published:** 2023-07-27

**Authors:** Natalia Potocka, Marzena Skrzypa, Maria Zadarko-Domaradzka, Zbigniew Barabasz, Beata Penar-Zadarko, Agata Sakowicz, Emilian Zadarko, Izabela Zawlik

**Affiliations:** 1Laboratory of Molecular Biology, Centre for Innovative Research in Medical and Natural Sciences, Medical College of Rzeszow University, Warzywna 1a, 35-959 Rzeszow, Poland; npotocka@ur.edu.pl (N.P.); mskrzypa@ur.edu.pl (M.S.); 2Institute of Physical Culture Sciences, Medical College of Rzeszow University, Cicha 2a, 35-959 Rzeszow, Poland; mzadarko@ur.edu.pl (M.Z.-D.); ezadarko@ur.edu.pl (E.Z.); 3Department of Physical Education, State University of Applied Sciences in Krosno, Rynek 1, 38-400 Krosno, Poland; zbigniew.barabasz@pans.krosno.pl; 4Institute of Health Sciences, Medical College of Rzeszow University, Kopisto 2a, 35-959 Rzeszow, Poland; bpenar@ur.edu.pl; 5Department of Medical Biotechnology, Medical University of Lodz, Żeligowskiego 7/9, 90-752 Lodz, Poland; agata.sakowicz@gmail.com; 6Institute of Medical Sciences, Medical College of Rzeszow University, Kopisto 2a, 35-959 Rzeszow, Poland

**Keywords:** *ADRB3*, Trp64Arg, polymorphism, anthropometric components, somatotype, cardiorespiratory fitness (CRF), oxygen uptake, oxygen pulse, physical activity

## Abstract

The *ADRB3* gene plays a role in energy expenditure by participating in lipolysis, which affects body composition and performance. The *ADRB3* rs4994 polymorphism has been studied in groups of athletes, overweight individuals, and obese and diabetic patients, but it has not been studied in young and healthy adults so far. In the present study, we examined the association of *ADRB3* rs4994 polymorphism with body composition, somatotype, cardiorespiratory fitness and physical activity in young, healthy adults (*N* = 304). All subjects had anthropometric measurements, and somatotypes were assessed using the Heath–Carter method. In addition, cardiorespiratory fitness and physical activity levels were assessed. Genotyping for the *ADRB3* gene was performed using a PCR-RFLP method. In the male group, body components were associated with the Trp64Trp genotype (waist circumference (*p* = 0.035), hip circumference (*p* = 0.029), BF (%) (*p* = 0.008), and BF (kg) (*p* = 0.010), BMI (*p* = 0.005), WHtR (*p* = 0.021), and BAI (*p* = 0.006)). In addition, we observed that the Trp64Trp genotype was associated with somatotype components (*p* = 0.013). In contrast, the Arg allele was associated with the ectomorphic components (0.006). We also observed a positive impact of the Trp64Trp genotype with maximal oxygen uptake (*p*= 0.023) and oxygen pulse (*p* = 0.024). We observed a negative relationship of the Trp64Trp genotype in the female group with reported moderate-intensity exercise (*p* = 0.036). In conclusion, we found an association of the Trp64 allele with anthropometric traits, somatotype and parameters describing physical performance in the male group. In the female subpopulation, we only found an effect of the polymorphism Trp64Arg on the level of physical activity for moderate-intensity exercise.

## 1. Introduction

In recent years, more and more research has focused on the influence of genes on fitness, not only in professional athletes but also in non-athletes. Genes influencing anthropometric characteristics and cardiorespiratory fitness are of great interest. Polymorphisms in these genes may contribute to the development of civilisation diseases, such as obesity or cardiovascular disorders. Increasingly, the impact of polymorphisms on undertaken physical activity, and not only on physical fitness, is being studied.

Included among the genes linked to anthropometric traits and performance are those encoding adrenergic receptors. Metabotropic receptors have been classified into two groups: α-adrenergic and β-adrenergic receptors. βARs are membrane receptors stimulated by neurotransmitters from the catecholamine group. They mediate the activation of adenylate cyclase induced by the action of G-protein, which is coupled to receptors, resulting in an increase in cAMP concentration in the cell. The βARs are functionally involved in cellular energy metabolism. Adrenergic receptors were divided into three groups: β1, β2, and β3. All these proteins show high homology in structure to each other. β1AR receptors are mainly found in cardiac muscle but also in the brain and kidneys. β2AR receptors are located in smooth muscle, the heart, skeletal muscle, white adipose tissue, and the liver. The β3AR receptors are mainly found in adipose tissue and are involved in thermogenesis and adipose tissue lipolysis. These receptors are crucial receptors mediating catecholamine-stimulated thermogenesis in adipose tissue. A decreased sensitivity of the βAR3 receptors was observed in obese individuals [1,2,3].

An important polymorphism involved in physical fitness genetics is rs4994 (T/C, Trp64Arg) in the *ADRB3* gene. The *ADRB3* gene is 3610 bp in length and is located on chromosome 8p11.23, containing two exons and one intron, encoding 408 amino acids. The rs4994 polymorphism involves a cytosine substitution in place of thymine at codon 64 of the gene encoding *ADRB3*, which has the effect of converting tryptophan to arginine (Trp64Arg) in the amino acid sequence [4]. The Trp64Arg polymorphism is functional and affects β3AR receptor activity by disrupting lipid metabolism through reduced lipolysis. The Trp64Arg polymorphism in the *ADRB3* gene affects the receptor affinity for norepinephrine, reducing lipolytic activity in adipocytes [5]. This polymorphism has been shown to affect body components such as fat distribution and BMI (body mass index). It also affects the possibility of losing weight, especially during physical exercise. Furthermore, the Trp64Arg polymorphism has been shown to be related to an increased risk of developing type 2 diabetes, obesity or hypertension [6,7,8,9]. A meta-analysis by Xie et al. found the Arg64 allele significantly increased the risk of overweight and obesity in children and adolescents, particularly in the East Asian population [10]. In addition, β3AR receptors affect cardiac function, and their stimulation reduces the force of myocardial contraction [11]. Moreover, it has been shown that β3AR receptors regulate angiogenesis and endothelium-dependent vascular relaxation in coronary microcirculation [12]. Furthermore, the Trp64Arg polymorphism has been analysed for its impact on physical performance, as it has been associated with some cardiovascular parameters. Kim et al. detected an association of the rs4994 polymorphism with the level of glucose and HDL-cholesterol in serum. They also observed that the allele distribution of the Trp64Arg polymorphism differed significantly according to the sports discipline practised [13]. Similar results have been obtained by Wessner et al., who reported the highest frequency of heterozygotes in athletes participating in team sports [14]. Santiago et al. observed significant differences between the prevalence of Trp64Arg and Arg64Arg genotype carriers in athletes compared to controls [15]. In addition, Arg64 allele carriers were more frequently recorded in endurance sports [16]. Therefore, rs4994 polymorphism in the *ADRB3* gene is increasingly considered a marker of athletic predisposition.

The significant effect of polymorphism in the *ADRB3* gene not only on anthropometric traits but also on performance and the previously published inconsistent results prompted us to undertake the current study on a homogeneous population. This study aimed to assess the prevalence of the rs4994 polymorphism in the *ADRB3* gene in a population of young, healthy individuals and to determine its impact on parameters describing body composition, cardiorespiratory fitness, and physical activity. The study of polymorphism in the *ADRB3* gene in a healthy young population is relevant to understand its impact on energy metabolism and health-related fitness. Additionally, the results of our study can be used as a reference for other studies on the development of obesity, as well as for studies on performance among professional athletes.

## 2. Materials and Methods

### 2.1. Study Group

The study involved students studying in major academic centres in Poland. The study group consisted of 304 healthy young adult subjects of Caucasian descent (149 women and 155 men). The criteria included negative exercise readiness interview after completing the Physical Activity Readiness Questionnaire, written consent to participate in the study and obtaining current medical examinations allowing participation in physical education classes. The exclusion criteria included a positive history of willingness to exercise (at least one positive answer in the Physical Activity Readiness Questionnaire), lack of consent to participate in the study and lack of current medical examinations, malaise before or during the exercise test. The average age in the female group was 20.3, and in the male group, 20.7 years. The cardiorespiratory fitness was evaluated based on a 20 m shuttle run test (20 m SRT). The maximum oxygen uptake (VO_2_max) was directly assessed using a K4b2 gas analyser (Cosmed, Roma, Italy) during the 20mSTR test. In addition, the subjects’ heart rates were recorded every five seconds throughout the test using a Polar Heart Rate Monitor (Polar Electro Oy, Kempele, Finland). Heart rate has also measured one minute and four minutes after cessation of exercise. In addition, all participants had their body composition assessed by bioimpedance using a Tanita scale (Tanita TBF 300, Tokyo, Japan), and body height was determined using a stadiometer. The body circumferences were measured using non-stretchable tape. The BMI (body mass index), WHR (waist to hip ratio), and WHtR (waist to height ratio) were calculated based on the results. Somatotype was calculated using the Heath–Carter method [17,18]. Physical activity level was measured using the Minnesota Leisure Time Physical Activity Questionnaire MLTPAQ. The questionnaire analysis included energy expenditure, expressed in kcal per week. The division of effort intensity into light (≤4 MET-metabolic equivalent), moderate (4.5–5.5 MET), and high intensity (≥6 MET) was also included [19,20]. A detailed description of the group selection and exclusion criteria has been published in the 2019 article [21]. The project was approved by the Bioethics Committee of the University of Rzeszow no. 20/12/2015. The overall characteristics of the study group are shown in Table 1.

### 2.2. Sample Collection and DNA Isolation

Saliva was collected using the GeneFiX DNA Saliva Collector kit containing stabilization buffer (Isohelix, Maidstone, UK). DNA was extracted using the GeneFiX DNA Saliva Isolation Kit according to the protocol of the producer (Isohelix, Maidstone, UK). DNA quality and quantity were assessed using a NanoDrop spectrophotometer (Thermo Scientific, Waltham, MA, USA). DNA samples had stored at −20 °C until use.

### 2.3. Genotyping of ADRB3

Polymorphism in the *ADRB3* gene was analysed using polymerase chain reaction (PCR)–restriction fragment length polymorphism (RFLP) assay. For PCR reaction, primers with the sequence forward F:5’AAT ACC GCC AAC ACC AGT G3’, reverse 5’GGT CAT GGT CTG AGT CTC G 3’ were used. PCR amplification reactions were carried out in a 20 µL volume containing 50–100 ng of DNA, 0.2 µM of each dNTP, 0.2 µM of each primer, 0.5 U Taq polymerase; 1x Pol buffer B with 2 mM MgCl_2_. Conditions used for the PCR reaction were as follows: 10 min initial denaturation at 95 °C, followed by 35 cycles at 94 °C for 30 s, 60 °C for 30 s, and 72 °C for 45 s; final extension was conducted at 72 °C for 7 min. The 150 bp amplification product was digested with the restriction enzyme MvaI (Thermo Scientific, Waltham, MA, USA). The reaction conditions and the length of the products obtained are shown in Table 2.

### 2.4. Statistical Analysis

Statistical analyses were performed using the Statistica 13.0 package (StatSoft, Krakow, Poland). The distribution of continuous variables was analysed using the Shapiro–Wilk test. Variables with a normal distribution were presented as means and standard deviations. These variables were analysed using the Student’s *t*-test. Variables with a non-normal distribution were presented as medians and quartile distributions and analysed using the Mann–Whitney U test. Nominal variables were presented as counts and percentages. Comparisons of nominal variables were made using the chi2 or chi2 test with Yates correction or Fisher’s exact test, depending on the abundance of the variables. Genotype frequencies were additionally verified against the Hardy–Weinberg equilibrium. Statistical analyses were carried out separately for the male and female groups.

## 3. Results

The genotype distribution and allele frequencies of Trp64Arg polymorphism in the ADRB3 gene are presented in Table 3.

The chi-square test showed no statistically significant deviations from the Hardy–Weinberg equilibrium for polymorphism 64 in the *ADRB3* gene, where *p* was 0.599.

The results of the relationship between the Trp64Arg polymorphism of the *ADRB3* gene and body components in the male and female groups are shown in Table 4. Statistical significance was reached only in the male cohort. There was a statistically significant association between the Trp64Trp genotype and the occurrence of higher values for the following parameters: waist circumference (80 cm vs. 78.7 cm; *p* = 0.035), hip circumference (99 cm vs. 96 cm; *p* = 0.029), fat content expressed percentage (14.9 vs. 13.2; *p* = 0.008), and kilograms (11.8 vs. 9.7; *p* = 0.010), as well as for BMI (23.8 vs. 22.3; *p* = 0.005), WHtR (0.45 vs. 0.435; *p* = 0.021), and BAI (23.3 vs. 21.5; *p* = 0.006).

Furthermore, we examined the association between the occurrence of the *ADRB3* rs4994 polymorphism and somatotype. We examined the components of somatotype in general and with a breakdown for each type separately. As a result, we found that men with the Trp64Trp genotype manifested more meso-endomorph features in their body type compared to carriers of the Arg allele. In addition, the Arg allele was shown to be strongly associated with typical ectomorphic body type (*p* = 0.006). The results are shown in Table 5.

Further, the analysis of cardiorespiratory fitness indicates that Trp64Trp homozygotes represent higher oxygen uptake (VO_2_(L/min)) and oxygen pulse (VO_2_/HR) values than the other genotypes, i.e., the VO_2_ was 4.15 vs. 3.79, (*p* = 0.023), and VO_2_/HR was 21.37 vs. 19.14 (*p* = 0.024), respectively. An association of cardiorespiratory parameters with the rs4994 *ADRB3* polymorphism was only found in the male group. All results of the relationship analyses between the *ADRB3* gene polymorphism and CRF parameters are presented in Table 6.

Moreover, we investigated the relationship between the rs4994 polymorphism in the *ADRB3* gene and physical activity levels. The results indicate that Trp homozygous females demonstrated lower scores in moderate-intensity exercise comparison to the carriers of the Arg allele, i.e., 185 vs. 258 (*p* = 0.036), respectively. The results of *ADRB3* gene polymorphism association with physical activity are shown in Table 7.

## 4. Discussion

As a result of our study, we found that there is a relationship between parameters describing body components and cardiorespiratory fitness and the Trp64Arg polymorphism in the *ADRB3* gene in healthy young adults. We also observed an association between the *ADRB3* gene polymorphism and the level of physical activity.

The results of the study indicate that there is an association between higher values for waist circumference, hip circumference or body fat percentage and body fat in kilograms, as well as higher values for BMI, WHtR and BAI, and the TT genotype (Trp64Trp), but only in the male subpopulation. The research by Matsushita et al. on a group of Japanese adults aged 40–69 years did not reach statistical significance for differences in BMI values. However, Trp64 carriers present higher BMI values than Arg64Arg homozygotes, which can be observed in the male group after accounting for age, lifestyle, physical activity and energy intake [22]. Similar results from a study conducted on a Japanese population over 40 years of age were obtained by Hara et al. In the results, slightly higher BMI values and a higher percentage of body fat can be seen in those with the Trp64Trp genotype compared to carriers of the Arg allele; however, these results did not reach statistical significance [23]. Different results from those presented in this paper were described in the study by Szendrei et al. on non-smoking and sedentary adults aged 18–50 years. The study population was subjected to a 6-month exercise programme and a change in dietary habits. Based on the results, they reported that carriers of the Arg64 allele had higher body fat mass, android fat (kg) and visceral adipose tissue (kg) than non-carriers of this allele. However, statistical significance was only reached in a group of females for body fat mass (kg). However, no differences were found in body weight and BMI [6]. Similar results, with no significant differences in BMI and waist circumference between genotypes, were reported by Dunajska et al. in a group of Polish postmenopausal women. Nevertheless, the Trp64/Arg64 genotype had been found to affect lower levels of high-density lipoprotein cholesterol (HDL-C) compared to women with the Trp64/Trp64 genotype [24]. In a study by Daghestani et al., in which the study population consisted of unrelated adult Saudi participants (ages 18 to 36 years), a higher prevalence of the Arg64 allele of the *ADRB3* gene was found in overweight and obese subjects compared to normal-weight subjects. In addition, carriers of the Arg64 allele had a higher waist and hip circumference and WHR versus those with the wild-type Trp allele. Moreover, these individuals had higher plasma levels of cholesterol, triglycerides, LDL, leptin, insulin and glucose compared to non-carriers of the Arg allele. These results show a significant association between the Trp64Arg polymorphism and the development of overweight and obesity in the study population [25]. Furthermore, a meta-analysis by Kurokawa et al., which included 44,883 individuals from East Asian and European populations, showed an association between BMI values and the Trp64Arg *ADRB3* polymorphism; Arg64 carriers had higher BMI. However, the difference in BMI was more pronounced in East Asian populations than among Europeans [26]. Although, it should be taken into account the allele distribution in the Asian population differs from that of the European population. The allele frequency in the European population is 93% for the Trp allele and 7% for the Arg allele, while it is 88% and 12% for the Asian population, respectively (data obtained from the NCBI database [27]). In addition, previous studies have not shown an association between the Trp64Arg variant and body composition phenotypes, at least in Caucasians [26,28]. In addition, in obese individuals, the Arg allele of the *ADRB3* gene was initially associated with an increased risk of weight gain [4]. A meta-analysis by González-Soltero et al. found that carriers of the Arg64 allele had a higher body mass index and elevated biochemical parameters, such as insulin and HDL-cholesterol levels, than Trp64Trp homozygotes. However, the Arg64 allele carriers responded to training and diet with improvements in anthropometric parameters. The presence of allele Arg64 has been found to influence the increased risk of developing cardiovascular disease [29]. Other studies have shown an increased BMI in allele Arg carriers, but only in sedentary individuals. On the contrary, BMI was not correlated with the *ADRB3* genotype in physically active individuals. Several studies have not observed an association between the Trp64Arg polymorphism and obesity phenotypes [13,30,31]. Previous studies have shown inconclusive results regarding the association between the *ADRB3* Trp64Arg polymorphism and body components and obesity. The discrepancies in results reported in the literature can partly be explained by differences in the selection of study groups that differed in ethnicity or gender. In addition, studies have been carried out on groups of healthy people with normal BMI and obese people with increased BMI, and people with diabetes have also been studied. Differences in the results may also result from the selection of people with different physical activities and lifestyles for the research groups. The reason for the inconsistent results regarding the association of the polymorphism with body components as well as with obesity may be the weak effect of the *ADRB3* gene polymorphism on fat gain and the pathogenesis of obesity.

In this study, we showed a statistically significant relationship between body type and the presence of the Try64Arg polymorphism in the *ADRB3* gene only in the male group. Individuals homozygous for Trp64Trp showed more mesomorphic and endomorphic characteristic components, while carriers of the Arg64 allele showed more ectomorphic components. It must be noted that both the endomorphic type and the mesomorphic type are characterized by a higher adipose tissue content than the ectomorphic type compared to the results obtained from the analyses on body components, namely higher values for the parameters describing adipose tissue content for the Trp64Trp genotype. Iglesia et al. investigated the effect of polymorphisms in various genes, including the *ADRB3* gene, on anthropometric variables and endurance performance. The study involved a group of cyclists with correct body weight, BMI and body fat. However, they did not show an association between polymorphismTrp64Arg and somatotype [32].

In this study, we showed that there was an association between the *ADRB3* gene polymorphism and parameters describing cardiorespiratory fitness. We showed that the Trp64Trp genotype influences the achievement of higher absolute oxygen ceiling and aerobic pulse values compared to individuals with other genotypes. This relationship was only observed in the male group. However, there is no conclusive information on the effect of the Trp64Arg polymorphism on oxygen ceiling and aerobic pulse. A study by Jesus et al. in a non-obese adolescent group from Brazil showed higher absolute oxygen ceiling values in individuals with the Trp64Trp genotype; however, these analyses did not reach statistical significance. In addition, they linked the presence of the Arg64 allele with lower values for maximal fat oxidation during aerobic exercise and higher LDL cholesterol concentrations [33]. The research by McCole et al. in a group of healthy postmenopausal women showed that VO_2_max values, whether expressed in litres per minute or millilitres per kilogram per minute, were significantly higher in heterozygotes compared to Trp64Trp homozygotes. However, this polymorphism was not associated with changes in any maximal or submaximal exercise cardiovascular hemodynamic responses [34]. In a study by Santiago et al., heterozygosity of a polymorphism in the *ADRB3* gene was reported to be associated with elite endurance performance. The study was carried out on a group of Spanish descent athletes (20–39 years) compared to a group of non-training healthy individuals (18–30 years) [15]. The results obtained by de Luis et al. in a group of obese patients showed that systolic blood pressure, glucose, triglycerides and insulin were significantly lower in the individuals with the Trp64Trp genotype, irrespective of diet. This indirectly affects exercise capacity through energy metabolism [35]. The association between *ADRB3* gene polymorphism and endurance performance can be explained by the numerous effects that β3AR receptors induce in the human tissues, including adipose tissue and the heart. β3AR receptors affect cardiac function. Their stimulation reduces the force of myocardial contraction. Furthermore, β3AR receptors regulate angiogenesis and endothelium-dependent vascular relaxation in the coronary microcirculation [11,36].

The rs4994 polymorphism in the *ADRB3* gene was also described in terms of the type of physical exercise undertaken. The distribution of genotypes and alleles of the Trp64Arg polymorphism in the *ADRB3* gene varied between disciplines. The Arg64 allele, for example, occurred at a higher frequency in those participating in sports such as volleyball and gymnastics, whereas a heterozygous genotype predominated in runners and cyclists [13,15]. We reported lower declared values for moderate-intensity efforts in the women group with the Trp64Trp genotype. In contrast, no such association was observed in the male group. Corbalan et al. showed that people with the Arg64 genotype had less physical activity. The study was carried out on a group of Spanish individuals between the ages of 20 and 60 and compared obese people with correct body weight individuals. They reported that young people with the Arg64 allele were more likely to be obese than those with other genotypes, regardless of gender and leisure-time physical activity [37]. Santiago et al. found that rs4994 polymorphism in the *ADRB3* gene also influences an individual’s sporting behaviour in strength sports, demonstrating that resistance exercise-based training is more suitable for carriers of the Arg64 allele [15]. In the future, research on the effect of the Trp64Arg polymorphism in the *ADRB3* gene on physical activity should be expanded.

The main limitation of our study was the group size. In the case of genetic polymorphism studies, the group size should be higher. However, our study group was well selected and defined, which is a strength of this study. Analyses of the relationship between the polymorphism and biochemical parameters, such as glucose or triglyceride levels, would add value to this study.

## 5. Conclusions

In conclusion, in the group of men, the Trp64 allele was found to have a negative effect on anthropometric characteristics and a positive impact on the parameters describing physical performance. Moreover, rs4994 polymorphism in the *ADRB3* gene showed an association with the somatotype. In contrast, we found no effect of the Trp64 allele in the female group on body composition traits and cardiorespiratory fitness. Females with the Trp64Trp genotype showed lower physical activity levels for moderate-intensity intensity.

## Figures and Tables

**Table 1 genes-14-01541-t001:** Characteristics of the study group.

	Male *N* = 155	Female *N* = 149
	Mean ± SD	Mean ± SD
Age (years)	20.7 ± 1.84	20.3 ± 1.24
Body weight (kg)	75.8 ± 12.07	59.3 ± 9.76
Body height (cm)	178 ± 6.79	165.3 ± 5.70
Waist circumference (cm)	80.8 ± 8.21	69.0 ± 6.80
Abdominal circumference (cm)	85.2 ± 9.40	75.1 ± 8.15
Hip circumference (cm)	98.4 ± 7.54	96.0 ± 7.44
BF (%)	15.4 ± 5.88	23.6 ± 7.01
BF (kg)	12.3 ± 6.70	14.6 ± 6.70
FFM (kg)	63.5 ± 6.59	44.6 ± 3.67
TBW (kg)	46.5 ± 4.82	32.7 ± 2.69
BMI (kg/m^2^)	23.9 ± 3.77	21.6 ± 3.16
WHtR	0.5 ± 0.05	0.4 ± 0.04
WHR	0.8 ± 0.04	0.7 ± 0.04
BAI	23.5 ± 3.58	27.2 ± 3.54
BSA	1.9 ± 0.15	1.6 ± 0.14
Distance (m)	1552.6 ± 430.90	953.6 ± 355.64
HRmax (bpm)	194.8 ± 8.39	189.6 ± 8.83
VO_2_max (ml/kg/min)	54.3 ± 8.02	44.8 ± 7.54
VO_2_ (L/min)	4.1 ± 0.62	2.6 ± 0.50
VO_2_/HR	20.9 ± 4.26	13.9 ± 2.89
HR after 1 min (bpm)	163.6 ± 12.88	161.4 ± 14.04
HR after 4 min (bpm)	125.6 ± 13.48	126.6 ± 13.79
HRmax—HR 1 min	31.2 ± 9.44	28.4 ± 10.97
HRmax—HR 4 min	69.2 ± 11.77	63.2 ± 12.74

BF—body fat, FFM—free fat mass, TBW—total body water, BMI—body mass index, WHR—waist to hip ratio, WHtR—waist to height ratio, BAI—body adiposity index, BSA—body surface area, HRmax—maximal heart rate, VO_2_max—maximal oxygen uptake, VO_2_—oxygen uptake, VO_2_/HR—oxygen pulse, HRmax—HR 1 min or 4 min—difference between HRmax and HR after 1 min or 4 min.

**Table 2 genes-14-01541-t002:** Restriction enzyme digestion reaction conditions and length of products obtained.

Gene/SNP	Enzyme	Restriction Fragment Lengths (bp)	Conditions Reaction
*ADRB3*/rs4994	MvaI	T—97, 27, 26 C—124, 26	Data 37 °C—3 h—1 U

The restriction digests products were subjected to electrophoresis using a 2% agarose gel for *ADRB3*. Midori Green (Nippon Genetics Europe GmbH, Düren, Germany) dye was used for visualisation.

**Table 3 genes-14-01541-t003:** Genotype distribution and allele frequency of the *ADRB3* gene rs4994 polymorphism.

*ADRB3*	Trp64Trp	Trp64Arg	Arg64Arg	Allele Trp	Allele Arg
All	246 (80.92)	54 (17.76)	4 (1.32)	546 (0.898)	62 (0.102)
Men	129 (83.23)	23 (14.84)	3 (1.94)	289 (0.906)	29 (0.094)
Women	117 (78.52)	31 (20.81)	1 (0.67)	265 (0.889)	33 (0.111)

Note: Data presented as *N* (%) for genotypes and *n* frequency for alleles.

**Table 4 genes-14-01541-t004:** Association of rs4994 polymorphism genotypes of the *ADRB3* gene with anthropometric parameters analysed.

	Male			Female		
	Trp64Trp (TT)	Carrier Arg64 (TC + CC)	*p*-Value	Trp64Trp (TT)	Carrier Arg64 (TC + CC)	*p*-Value
Waist circumference (cm)	80 (75.5–85)	78.7 (74.5–80.0)	0.035 *	68.5 (65–72)	67 (65–70.3)	0.545
Hip circumference (cm)	99 (93.5–104)	96 (92–99)	0.029 *	96 (90–100)	93.5 (91–99)	0.806
Abdominal circumference (cm)	84 (79–91)	82 (79–85)	0.074	75 (70.5–79)	72.5 (70.25–76.75)	0.280
BF (%)	14.9 (11.5–19.5)	13.2 (9.6–14.3)	0.008 *	23.3 (18.8–28.3)	23.2 (19.8–26.9)	0.967
BF (kg)	11.8 (8.4–16)	9.7 (6.4–10.8)	0.010 *	13.6 (10.6–18.8)	13 (10–16.7)	0.873
FFM (kg)	63.8 (59.1–69.5)	62.1 (59.13–65.5)	0.195	44.30 (42.4–46.3)	44.20 (42.35–46.6)	0.915
TBW (kg)	46.7 (43.3–50.9)	45.45 (43.3–48)	0.196	32.40 (31–33.9)	32.40 (31–34.15)	0.918
BMI	23.8 (21.8–26.3)	22.3 (20.6–23.5)	0.005 *	21.5 (19.6–23.3)	21 (19.6–22.3)	0.080
WHR	0.81 (0.79–0.85)	0.81 (0.8–0.83)	0.499	0.72 (0.7–0.74)	0.72 (0.7–0.74)	0.799
WHtR	0.45 (0.42–0.49)	0.435 (0.41–0.45)	0.021 *	0.41 (0.39–0.44)	0.41 (0.39–0.43)	0.678
BAI	23.3 (21.3–26.2)	21.5 (19.5–23.4)	0.006 *	26.7 (21.1–29.6)	27.6 (24.6–28.7)	0.857
BSA	1.90 (1.8–2.0)	1.90 (1.8–2.0)	0.261	1.6 (1.6–1.7)	1.6 (1.55–1.7)	0.817

BF—body fat, FFM—free fat mass, TBW—total body water, BMI—body mass index, WHR—waist to hip ratio, WHtR—waist to height ratio, BAI—body adiposity index, BSA—body surface area. Note: Data are presented as median (minimum: maximum); *p*-values were calculated using the Mann–Whitney U test, * significant at *p* < 0.05.

**Table 5 genes-14-01541-t005:** Association between the rs4994 polymorphism in the *ADRB3* gene and somatotype.

	Male			Female		
	Trp64Trp (TT)	Carrier Arg64 (TC + CC)	*p*-Value	Trp64Trp (TT)	Carrier Arg64 (TC + CC)	*p*-Value
Somatotype	4 (4.7–8)	8 (4.3–9.2)	0.013 *	3 (2–11)	7 (2.5–11)	0.233
Endomorphic	2.4 (1.9–3.5)	2.5 (2.0–2.9)	0.555	3.4 (2.8–4.1)	3 (3.2–3.9)	0.744
Mesomorphic	4 (3–5.1)	3.4 (2.6–4.2)	0.056	2.9 (1.8–3.5)	3 (1.9–4)	0.588
Ectomorphic	2.3 (1.1–3.3)	3.2 (2.4–3.9)	0.006 *	2.5 (1.6–3.5)	2.8 (1.8–3.5)	0.837

Note: Data are presented as median (minimum: maximum); *p*-values were calculated using the Mann–Whitney U test, * significant at *p* < 0.05.

**Table 6 genes-14-01541-t006:** Association of *ADRB3* gene polymorphism with the cardiorespiratory fitness parameters analysed.

	Male			Female		
	Trp64Trp (TT)	Carrier Arg64 (TC + CC)	*p*-Value	Trp64Trp (TT)	Carrier Arg64 (TC + CC)	*p*-Value
Distance (m)	1560 (1260–1880)	1430 (1210–1800)	0.608	920 (660–1220)	860 (670–1090)	0.051
HRmax (bpm)	194 (189–200)	196 (193–202.75)	0.241	191 (185–196)	189 (178.5–196)	0.404
VO_2_max (ml/kg/min)	53.8 (48.2–60.10)	53 (49.4–57.03)	0.769	44.55 (40.18–48.7)	44.39 (42.2–49.7)	0.954
VO_2_ (L/min)	4.15 (3.60–4.57)	3.79 (3.44–4.23)	0.023 *	2.56 (2.21–2.95)	2.56 (2.39–2.92)	0.759
VO_2_/HR	21.37 (18.51–23.34)	19.14 (17.75–21.72)	0.024 *	16.03 (14.04–18.3)	15.38 (14.29–17.71)	0.682
HR difference after 1 min	31 (24–38.25)	30 (27–32.75	0.465	27.5 (21–36)	24 (19–30)	0.074
HR difference after 4 min	69.5 (62–78)	70.5 (60.25–8.5)	0.794	63 (57–69.25)	60 (51–65)	0.075

HRmax—maximal heart rate, VO_2_max—maximal oxygen uptake, VO_2_—oxygen uptake, VO_2_/HR—oxygen pulse. Note: Data are presented as median (minimum: maximum); *p*-values were calculated using the Mann–Whitney U test, * significant at *p* < 0.05.

**Table 7 genes-14-01541-t007:** Association of rs4994 polymorphism genotypes of the *ADRB3* gene with physical activity levels.

	Male			Female		
	Trp64Trp (TT)	Carrier Arg64 (TC + CC)	*p*-Value	Trp64Trp (TT)	Carrier Arg64 (TC + CC)	*p*-Value
Minnesota low intensity	455 (194–738)	410 (300–987)	0.632	404 (184–637)	429 (225–581)	0.799
Minnesota moderate intensity	295 (62–647)	215 (78–414)	0.466	185 (42–472)	258 (152–533)	0.036 *
Minnesota high intensity	891 (249–1846)	796 (171–1848)	0.835	385 (118–869)	334 (164–757)	0.741
Minnesota total score	1931 (984–3241)	202 (856–2832)	0.644	1126 (634–2086)	1146 (908–1619)	0.689

Note: Data are presented as median (minimum: maximum); *p*-values were calculated using the Mann–Whitney U test, * significant at *p* < 0.05.

## Data Availability

The data presented in this study are available upon request from the corresponding authors.

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
