# Peer review of "Effects of the Trp64Arg Polymorphism in the ADRB3 Gene on Body Composition, Cardiorespiratory Fitness, and Physical Activity in Healthy Adults"

_genes, 2023, doi:10.3390/genes14081541_

Round 1

Reviewer 1 Report

In this study the authors looked for correlations between rs4994 polymorphism and some body and physical activities in young men and women. The objective is not new at all, but necessary due to contradictory previous results. The paper is very well written, with a clear exposition of results. 

Some important issues must be improved.

- Line 23: Begin the abstract with a justification of the election of the gene.

- Line 25: Indicate n.

- Line 65: ref 1 is not appropriated here, it refers to sport.

- Line 74: ref 1 is not appropriated.

- Line 78: ref 8 and 9 are not appropriated, they do not conclude influence of this polymorphism.

- Line 90: It is the same than ref 8, here it is apropriated.

- Line 92: here ref 1 is appropiated.

- Line 97: A comment about utility of the study is needed.

- Line 98: This paragraph comes from instructions. delete.

- Study group: Indicate minimun sample size calculations.

- Line 119: Define abreviations.

- Line 122: Descriptions of physical tests performed by subjects and physical activity level classification are needed.

- Table 1: Delete Title 2.

- Table 4: Indicate that significant p-values are in bold in all tables.

- Line 194: Define p UM-W.

- Line 201: Have you studied statistics between one T allele against the rest? It can be interesting.

- Table 6: ADRB3 in italics.

- Discussion: Please clasify the references with and without correlation in the same direction than this paper. And indicate the characteristics of the sample, I mean age, physical activity, health state, ethnicity. Perhaps you can find explanations for the contradictions then.

- Line 263: These papers are not "later" (revise english), they were published before 25 and 26.

- Line 268: Revise english.

- Line 304: This study was performed in adolescents, indicate.

- Line 324: Revise english.

- Line 408: It lacks complete citation.

- Line 411: Ref repeated: it is ref 8.

- Line 421: Incomplete ref.

- Line 440: Indicate doi instead.

English is good in average, please see some indications.

Reviewer 2 Report

Table 1 has some typo errors.

Male/Title , waist circum is reported 2 times with different values each time

line 98-106 must be deleted.

line 115: what is a fix polar heart belt ? is clear it is fix.

line 122: inclusion and exclusion criteria must be reported.

in any tables, r values for correlation must be reported. p values are not  enough.

limitations should be placed at the end of the discussion.

Round 2

Reviewer 1 Report

Please add paragraph about minimun sample size calculations in MM.